# Abscisic Acid and Its Receptors LANCL1 and LANCL2 Control Cardiomyocyte Mitochondrial Function, Expression of Contractile, Cytoskeletal and Ion Channel Proteins and Cell Proliferation via ERRα

**DOI:** 10.3390/antiox12091692

**Published:** 2023-08-30

**Authors:** Sonia Spinelli, Lucrezia Guida, Mario Passalacqua, Mirko Magnone, Vanessa Cossu, Gianmario Sambuceti, Cecilia Marini, Laura Sturla, Elena Zocchi

**Affiliations:** 1Laboratorio di Nefrologia Molecolare, IRCCS Istituto Giannina Gaslini, Via Gerolamo Gaslini 5, 16147 Genova, Italy; 2Section Biochemistry, Department of Experimental Medicine (DIMES), University of Genova, Viale Benedetto XV, 1, 16132 Genova, Italy; l.guida@unige.it (L.G.); mario.passalacqua@unige.it (M.P.); mirko.magnone@unige.it (M.M.); 3Section Human Anatomy, Department of Experimental Medicine (DIMES), University of Genova, 16126 Genova, Italy; vanessa.cossu@edu.unige.it; 4U.O. Medicina Nucleare, IRCCS Ospedale Policlinico San Martino, 16131 Genova, Italy; sambuceti@unige.it (G.S.); cecilia.marini@unige.it (C.M.); 5Department of Health Sciences, University of Genoa, 16132 Genova, Italy; 6Institute of Molecular Bioimaging and Physiology (IBFM), National Research Council (CNR), 20100 Milan, Italy

**Keywords:** mitochondrial function, cell cycle, cardiomyocyte functional proteins, proton leak

## Abstract

The cross-kingdom stress hormone abscisic acid (ABA) and its mammalian receptors LANCL1 and LANCL2 regulate the response of cardiomyocytes to hypoxia by activating NO generation. The overexpression of LANCL1/2 increases transcription, phosphorylation and the activity of eNOS and improves cell vitality after hypoxia/reoxygenation via the AMPK/PGC-1α axis. Here, we investigated whether the ABA/LANCL system also affects the mitochondrial oxidative metabolism and structural proteins. Mitochondrial function, cell cycle and the expression of cytoskeletal, contractile and ion channel proteins were studied in H9c2 rat cardiomyoblasts overexpressing or silenced by LANCL1 and LANCL2, with or without ABA. Overexpression of LANCL1/2 significantly increased, while silencing conversely reduced the mitochondrial number, OXPHOS complex I, proton gradient, glucose and palmitate-dependent respiration, transcription of uncoupling proteins, expression of proteins involved in cytoskeletal, contractile and electrical functions. These effects, and LANCL1/2-dependent NO generation, are mediated by transcription factor ERRα, upstream of the AMPK/PGC1-α axis and transcriptionally controlled by the LANCL1/2–ABA system. The ABA-LANCL1/2 hormone-receptor system controls fundamental aspects of cardiomyocyte physiology via an ERRα/AMPK/PGC-1α signaling axis and ABA-mediated targeting of this axis could improve cardiac function and resilience to hypoxic and dysmetabolic conditions.

## 1. Introduction

Abscisic acid is a terpenoid plant and animal hormone, with a cross-kingdom conserved role as a stress hormone, regulating cell responses to stimuli as diverse as root water availability and blood glucose levels [1]. The origin of ABA dates back to unicellular algae and bacteria and its conservation in modern, far more complex organisms testifies to its important role in cell and species conservation, allowing adaptation to changing environmental conditions.

ABA and its mammalian receptors LANCL1 and LANCL2 were recently shown to play a hitherto unrecognized role in the response of cardiomyocytes to hypoxia. The heart is a very well-perfused organ and the necessity to endure severe hypoxia does not occur under physiological conditions, which is not the case with skeletal muscle, where extreme physical exertion can shift the metabolism from the aerobic to the anaerobic condition. Indeed, type II skeletal myocytes are especially adapted to fast and brief bursts of contractile activity that are largely oxygen and mitochondria independent, while type I skeletal myocytes rely chiefly on the oxidative metabolism for energy production during prolonged physical exercise. The cardiac muscle is not as well adapted to hypoxia as the skeletal muscle; cardiomyocytes have an obligate aerobic metabolism and are thus exposed to the damaging effects of sudden oxygen deprivation.

One of the fundamental players in heart protection from hypoxia-derived damage is the gaseous hormone nitric oxide (NO), also an evolutionarily ancient signal molecule. Indeed, NO protects cardiomyocyte function not just after hypoxia/reoxygenation (a time-limited pathological event), but constitutively, under physiological conditions, improving electrical transmission, contractility, energy metabolism and myocyte growth [2]. Indeed, NO deficiency is associated with heart diseases [3] and NO administration can improve cardiac performance [4]. In the rat cardiomyocyte cell line H9c2, hypoxia induces the production of endogenous ABA and the ABA-LANCL1/2 hormone-receptor system activates a signaling pathway involving AMPK and PGC-1α, which leads to a concerted series of transcriptional and post-transcriptional events leading to increased NO generation, the expression and phosphorylation of eNOS, transcription of the enzyme GTPCH, which synthesizes the coenzyme tetrahydrobiopterin (TBH4), necessary for NO generation, and of the arginine transporter CAT-2A, required for arginine entry into the cells [5]. Interestingly, LANCL1/2-overexpressing H9c2 exposed to hypoxia shows a significantly higher mitochondrial proton gradient (ΔΨ) as compared with LANCL1/2-silenced cells, and the ΔΨ further increases after reoxygenation. In addition, treatment with ABA improves the mitochondrial ΔΨ in reoxygenated LANCL1/2-overexpressing cells, an effect which is almost completely lost in double-silenced cells. These observations suggest a role for the ABA-LANCL1/2 system in the maintenance of mitochondrial ΔΨ and consequently in the generation of metabolic energy in cardiomyocytes.

Mitochondrial function is of the utmost importance not only for cardiomyocyte survival of hypoxia, a condition that they do not normally experience, but for the contractile and electrical functions of cardiomyocytes under normoxic conditions. We hypothesized that the ABA/LANCL1-2 system, by increasing mitochondrial ΔΨ under normoxia, as well as after hypoxia/reoxygenation, could increase energy production in cardiomyocytes, improving the structural and metabolic features of the cells, in a word, improve cardiomyocyte “fitness”. In cardiac myocytes, mitochondrial function and biogenesis are controlled by the PGC-1α/ERRα team of transcription factors [6,7,8]. In skeletal myocytes, the ABA-LANCL1/2 system controls mitochondrial function, increasing mitochondrial-DNA content and respiration via a signaling pathway involving AMPK and PGC-1α [9] and PGC-1α together with ERRα have been shown to control transcription of several genes critical for mitochondrial-energy production in cardiac and skeletal muscle in vivo [7]; moreover, both PGC-1α and ERRα are transcriptionally upregulated in LANCL1/2-overexpressing human brown and beige adipocytes [10]. Starting from these observations, a role for ERRα in the mitochondrial effects of the ABA/LANCL1-2 system in cardiomyocytes can be hypothesized.

The aims of this study were two-fold: (i) to compare mitochondrial number, glucose- and palmitate-dependent respiration, OXPHOS uncoupling, expression of cytoskeletal, contractile and ion channel proteins, cell morphology and doubling time in rat H9c2 cardiomyocytes overexpressing, or silenced by, LANCL1 and LANCL2, cultured in the presence or in the absence of nanomolar ABA; (ii) to investigate whether ERRα is involved in the signaling pathway activated by the ABA/LANCL system in cardiomyocytes.

## 2. Materials and Methods

### 2.1. Cell Culture

The rat embryonic cardiomyocyte H9c2 cell line was purchased from ATCC (LGC Standards s.r.l. Milan, Italy) and was cultured in high glucose DMEM (Sigma-Aldrich, Milan, Italy) containing 10% fetal bovine serum (Sigma-Aldrich, Milan, Italy), penicillin (62.5 μg/mL) and streptomycin (100 μg/mL) (Sigma-Aldrich, Milan, Italy). Cells were kept at 37 °C in a humidified atmosphere with 5% CO_2_.

### 2.2. Lentiviral Cell Transduction

The lentiviral plasmids pLV[shRNA]-Puro-U6 encoding for a control scramble shRNA (SCR), for the shRNA against rat LANCL1 (SHL1), for the shRNA against rat LANCL2 (SHL2) and for the shRNA targeting rat ERRα (SHERRα) (plasmid ID: VB010000-0005mme, VB181016-1107sen, VB181016-1124zjp, VB221005-1073jxq), were obtained from Vector Builder (Chicago, IL, USA). hLANCL1 (OVL1) and hLANCL2 (OVL2) were overexpressed in rat H9c2 cardiomyocytes with pBABE vectors, constructed as described in [9], with the empty vector pBABE (Addgene, Watertown, MA, USA) as negative control (PLV). The method for lentiviral transduction is described in [9].

### 2.3. qPCR Analysis

H9c2 cells were incubated with or without 100 nM ABA or 100 μM L-NAME for 4 h after being serum starved for 18 h. Total RNA was extracted from cardiomyocytes with the RNeasy Micro Kit (Qiagen, Milan, Italy). cDNA was obtained from 1 μg of total RNA by using iScript cDNA Synthesis Kit (Bio-Rad, Milan, Italy) and qPCR reactions were performed in an iQ5 Real-Time PCR detection system (Bio-Rad, Milan, Italy) as described in [10]. All primers were designed using Beacon Designer 2.0 software (Bio-Rad, Milan, Italy), and their sequences are listed in Appendix A. Values were normalized on hypoxanthine-guanine phosphoribosyltransferase-1 (Hprt1) mRNA expression. Statistical analysis of the qPCR was obtained with the iQ5 Optical System Software version 1.0 (Bio-Rad Laboratories, Milan, Italy) by 2^−△△Ct^ method [9]. Absence of nonspecific PCR products was checked by analyzing the dissociation curve of each amplification cycle.

### 2.4. Western Blot

H9c2 rat cardiomyocytes (1 × 10^6^/well) were seeded in 6-well plates in DMEM with 10% FBS and 1% penicillin/streptomycin. After serum deprivation for 18 h, cells were washed once in Krebs-Ringer HEPES buffer (KRH) and then incubated in KRH with 5 mM glucose for 60 min at 37 °C with or without 100 nM ABA. The supernatant was removed and cells were scraped in 200 μL of 20 mM Tris-HCl pH 7.4, 150 mM NaCl, 1 mM EDTA and 1% NP40 containing a protease inhibitor cocktail. After brief sonication, the protein content was measured on an aliquot of each lysate. Lysates (25 μg) were loaded on 10% polyacrylamide gel, proteins were separated by SDS-PAGE and transferred to nitrocellulose membranes (Bio-Rad, Milan, Italy). Membranes were saturated by incubation for 1 h with TBST containing 5% non-fat dry milk and further incubated for 1 h at room temperature with the primary antibodies (Appendix A). After incubation with the appropriate secondary antibodies (Appendix A) and ECL detection (GE Healthcare, Milan, Italy), band density was estimated with the ChemiDoc imaging system (Bio-Rad, Milan, Italy).

### 2.5. Glucose Transport Assays

Rat H9c2 cells infected with the empty vector (PLV), overexpressing hLANCL1 and hLANCL2 (OVL1+2), infected with a scramble shRNA (SCR) or silenced for the expression of both rLANCL1 and rLANCL2 (SHL1+2) were cultured overnight at 1 × 10^4^/well in a 96-well plate in DMEM (5 mM glucose) without serum. Cells were washed once with DMEM and then incubated for 30 min at 37 °C in DMEM without (controls) or with 100 nM ABA. Cells were then washed with KRH at 37 °C and 50 μM of the fluorescent glucose analog 2-NBDG was added. After 10 min, the supernatant was removed, wells were washed once with ice-cold KRH, 50 μL KRH was added to each well, and the mean fluorescence (excitation at 480 nm and emission at 540 nm) from 9 acquisitions/well was calculated. Each experimental condition was assayed in at least 6 wells. Unspecific 2-NBDG uptake was subtracted from each experimental value as reported in [11].

### 2.6. JC-1 Analysis

Cardiomyocytes were stained with the cationic dye JC-1 (ThermoFisher Scientific, Waltham, MA, USA), with the same method already described in [5]. Briefly, H9c2 cells were seeded at 3 × 10^4^ onto µ-slide wells, treated or not with 4 µM CsA for 2 h, incubated with JC-1 (2.5 µg/mL) for 20 min at 37 °C in a 5% CO_2_ incubator and then imaged live. The red/green ratio, expression of the mitochondrial proton gradient [12], was analyzed after a background subtraction with the ImageJ software (v1.8.0, National Institutes of Health, Bethesda, MD, USA), using quantitative analysis of the fluorescence of selected ROIs.

### 2.7. Mitochondrial and Cytoskeletal Staining

H9c2 cardiomyocytes overexpressing or silenced by the expression of both LANCL proteins were plated on glass coverslips and then stained with MitoTracker^TM^ Deep Red FM (ThermoFisher, Waltham, MA, USA). After preparing the solution as previously described in [10], H9c2 cells were stained for 40 min at 37 °C in a 5% CO_2_ incubator and then imaged live. Moreover, the actin cytoskeleton (F-actin) was visualized using Phalloidin Alexa Fluor 488 (diluted 1:20, Cell Signaling Technology, Danvers, MA, USA). Rat H9c2 cardiomyocytes, cultured as mentioned above, were also set in 4% paraformaldehyde and permeabilized with 0.1% Triton X-100 (Sigma-Aldrich, Milan, Italy). Detection of cytoskeleton staining was performed with the following antibodies: mouse anti-α-tubulin (diluted 1:1000 in PBS, Sigma-Aldrich, Milan, Italy), rabbit anti β-catenin (diluted 1:100 in PBS, Cell Signaling Technology, Danvers, MA, USA), mouse anti-MYH7 (diluted 1:50, Santa Cruz Biotechnology Inc., Dallas, TX, USA) and mouse anti-Cx43 (diluted 1:100, Santa Cruz Biotechnology Inc., Dallas, TX, USA). The slides were rinsed in PBS and mounted with a ProLong Gold antifade mountant (ThermoFisher Scientific, Waltham, MA, USA). Images were acquired on a Leica TCS SP confocal laser scanning microscope, equipped with 476, 488, 543 and 633 excitation lines with a 60× Plan Apo oil objective. Fluorescence signals were then analyzed, after background subtraction, by ImageJ software (v1.8.0, National Institutes of Health, Bethesda, MD, USA) using a semi-manual method based on the delimitation of the cells. Three fields were chosen at random in two slide preparations for each sample.

### 2.8. Seahorse Analysis

Oxygen consumption rate (OCR) and extracellular acidification rate (ECAR) were measured with a Seahorse XFp extracellular flux analyzer (Agilent Technologies, Santa Clara, CA, USA). H9c2 cells were seeded in XF plates at a density of 6000 cells/well 24 h prior to analysis. Cells were then incubated at 37 °C for 45 min without CO_2_ with the Agilent Seahorse DMEM pH 7.4, containing glucose (25 mM), glutamine (2 mM) and pyruvate (1 mM). The bioenergetic profile was measured using the Cell Mito Stress Test Kit (Cat. #103010-100). Three measurements of OCR and ECAR were initially performed without additions to the cells, to obtain basal respiration and basal ECAR, and then repeated after the sequential injections of oligomycin (1.5 µM, ATP synthase inhibitor, enables the calculation of ATP-linked respiration), carbonyl cyanide-4-(trifluoromethoxy) phenylhydrazone (FCCP, 2.0 µM, proton gradient dissipator, enables the calculation of the maximal respiration rate) and rotenone (0.5 µM, respiratory Complex I inhibitor) plus antimycin A (0.5 µM, respiratory Complex III inhibitor), which completely blocks respiration and enables the calculation of non-mitochondrial oxygen consumption. OCR (pmolO_2_/min) and ECAR (mpH/min) were normalized to the total cell number determined directly in the plate after each experiment. To determine the intrinsic rate and capacity of cells to oxidize palmitate in the absence or limitation of other exogenous substrates, the XF Palmitate Oxidation Stress Test Kit (Cat. #103693-100) was used. Briefly, H9c2 cardiomyocytes were incubated at 37 °C for 45 min without CO_2_ with the Agilent Seahorse DMEM pH 7.4, containing 1 mM sodium palmitate/0.17 mM BSA solution (Palm-BSA), 25 mM glucose, 2 mM glutamine and 1 mM pyruvate. The OCR was monitored upon serial injections of etomoxir (4 μM, an inhibitor of long chain fatty acid oxidation) or DMEM, oligomycin (2 μM), FCCP (1 μM) and a rotenone/antimycin A mixture (1 μM). Whole-cell OCR was normalized to the final cell number as determined by manual cell counting.

### 2.9. Determination of Intracellular NAD^+^ and ATP Levels

To determine the content of ATP and NAD^+^, H9c2 were seeded in 6-well plates in DMEM supplemented with 10% FBS. At confluence, the cells were washed in PBS buffer, harvested and deproteinized with 5% TCA. After centrifugation (700× *g* for 30 s), the supernatants were neutralized by removing TCA with diethyl ether and were analyzed by HPLC [13]. NAD^+^ and ATP values were normalized to protein concentrations [14].

### 2.10. Cell Volume Measurement

To determine cell volume, H9c2 cardiomyocytes overexpressing or silenced by the expression of both LANCL proteins were treated with a solution of 1.25 μg/mL calcein-AM (Sigma-Aldrich, Milan, Italy) in isotonic PBS for 30 min at 37 °C and then imaged live. Z-stack imaging series were acquired using a Leica TCS SP2 confocal microscope and a 60× oil objective with a numerical aperture of 1.4 and a pinhole size of 1 Airy unit. To reduce the time for acquisition and to minimize photobleaching, scanning was performed at the fastest scan speed possible. Imaging emission and detection settings were adjusted to an optimal level, minimizing pixel saturation for each channel. The offset and gain of the system were also utilized to minimize background signal. Once imported as TIFF images in FIJI ImageJ, image stacks were then separated into the independent channels, filtered to emphasize the borders and thresholded automatically using the Otsu method from the automatic threshold dropdown box. A region of interest was then selected around each cell and an open-source plugin integrated imaging platform FIJI ImageJ (https://visikol.com/blog/2018/11/29/blog-post-loading-and-measurement-of-volumes-in-3d-confocal-image-stacks-with-imagej (accessed on 18 April 2023)) was used to calculate the number of voxels in the ROI in the image stack [15,16]. Briefly, the plugin calculates the area of each individual ROI of each section, then sums the areas and multiplies the sum by the depth of each individual section, then shows the outcome.

### 2.11. DNA Extraction and Determination of Mitochondrial Levels by qPCR Analysis

Total DNA was extracted from cardiomyocytes (1 × 10^6^ cells on 6-cm plates) with the QIAamp DNA Micro Kit (Qiagen, Milan, Italy). DNA quantity and purity were evaluated with the NanoDrop 1000 (Thermo Fisher Scientific, Waltham, MA, USA). The mitochondrial-to-genomic DNA ratio was determined by qPCR analysis using specific primers (Appendix A) for mitochondrial (MT-ND1) and nuclear (Hprt1) transcripts, as described in [10].

### 2.12. Statistical Analysis

Results were expressed as mean ± SD. Statistical analysis was performed using the GraphPad Prism 7 Software (GraphPad Software Inc., Boston, MA, USA). Comparisons were drawn from an unpaired, two-tailed Student’s *t*-test, if not otherwise indicated. Statistical significance was set at *p* < 0.05.

## 3. Results

### 3.1. Mitochondrial Number, Proton Gradient, Proton Leak and Respiration Are Increased in LANCL1/2-Overexpressing Cells, Whereas a Decrease Is Observed in LANCL1/2-Silenced H9c2 Cardiomyocytes

It has been previously reported that, by activating the AMPK/PGC-1α axis and Akt, the ABA-LANCL1/2 system controls primary mechanisms in the response of cardiomyocytes to hypoxia/reoxygenation. In order to understand to what extent LANCL proteins affect the oxidative metabolism and mitochondrial respiration under physiological, normoxic conditions, LANCL1 and LANCL2 were simultaneously silenced or overexpressed in H9c2 cells. shRNA-mediated silencing (SHL1+2) reduced mRNA and protein levels by approximately 80% in cardiomyocytes compared with the control cells infected with a scrambled shRNA (SCR) (Figure 1A).

The overexpression of LANCL1 and LANCL2 (OVL1+2) was obtained by retroviral infection and was confirmed by Western blot analysis. All experiments were performed on cells expressing approximately 8- and 30-times higher levels of LANCL1 and LANCL2, respectively, than control cells infected with the empty vector (PLV) (Figure 1B). To evaluate whether LANCL1/2-overexpressing cells possessed increased respiratory capacity compared with cells double-silenced for both proteins, mitochondrial proton gradient and respiration were analyzed. No significant differences in any of the parameters explored were observed between cells infected with the empty vector (PLV) used for LANCL1/2-overexpression and cells transfected with the scrambled sequences (SCR) used for LANCL1/2 silencing, indicating that neither vector per se affected mitochondrial number or respiratory activity (Appendix A, panel A). In LANCL1/2-overexpressing H9c2 loaded with the mitochondria-specific fluorescent dye MitoTracker, the total mitochondrial fluorescence showed an approximate three-fold increase compared with LANCL1/2 double-silenced cells (Figure 1C). In addition, the parameters of circularity and solidity, reflecting mitochondrial morphological but also functional changes, were both significantly decreased in LANCL1/2-overexpressing vs. -silenced cells. Increased cardiac mitochondrial circularity (a value closer to 1.0, as observed in the double-silenced cells, Figure 1C, right panel) has been linked to ischemia/reperfusion and radical-induced stress [17]. Low solidity (a value closer to 0, as observed in overexpressing cells, Figure 1C, right panel) describes mitochondria which are not uniform in shape and are more subject to branching and subsequent fission [18], possibly indicating a higher tendency to mitochondrial biogenesis in overexpressing cells.

A higher mitochondrial number in LANCL1/2-overexpressing cells also resulted in increased cell respiration. The oxygen consumption rate (OCR) was measured using the Seahorse XFp analyzer, with the sequential addition of oligomycin (an inhibitor of the ATP synthase proton channel, which allows the ATP-linked respiration to be calculated), FCCP (a proton transporter, which completely dissipates the mitochondrial proton gradient, allowing the calculation of the maximal respiration rate) and rotenone/antimycin A (inhibitors of the electron transfer chain, which completely inhibit mitochondrial respiration, allowing the measurement of baseline, non-mitochondrial, oxygen consumption). Basal, ATP-linked and maximal respiration rates all significantly increased in LANCL1/2-overexpressing cells, but not by the same proportion: the basal rate doubled and the maximal tripled in overexpressing vs. silenced cells (Figure 1D, upper left panel). The spare respiratory capacity (SRC), i.e., the ratio between maximal and basal respiration and a measure of mitochondrial capacity to adapt to stress conditions, was consequently two-fold higher in LANCL1/2-overexpressing vs. double-silenced cells (approx. four vs. two). The extracellular acidification rate (ECAR) was, however, similar in overexpressing and in double silenced cells (Figure 1D, upper right panel), resulting in a significantly higher OCR-to-ECAR ratio in LANCL1/2-overexpressing vs. double silenced H9c2 (Figure 1D, lower left panel). Finally, the percentage of the OCR dependent on fatty acid oxidation (inhibited by etomoxir) was higher in overexpressing vs. double-silenced cells for both the basal and maximal respiration rates (approx. two- and seven-times higher, respectively), but it was similar in the two cell types for the ATP-linked respiration (Figure 1D, lower right panel), indicating that the higher fatty acid-dependent OCR in overexpressing cells was not utilized for ATP generation. According to these results, the overexpression of LANCL1/2 significantly increases, whereas their combined silencing reduces, mitochondrial number, basal, maximal and fatty acid-dependent respiration rates in cardiomyocytes. Interestingly, the mitochondrial “proton leak” (calculated as the difference between basal and ATP-linked respiration) was also two-fold higher in LANCL1/2-overexpressing compared with double-silenced cells (Figure 1D).

To directly investigate the magnitude of the ∆Ψ in LANCL1/2-overexpressing or -silenced cardiomyocytes, we used the ∆Ψ-sensitive ratiometric dye JC-1. This fluorescent molecule accumulates within mitochondria and changes its emission from green to red as the ∆Ψ increases [12]. As shown in Figure 1E, left panel, mitochondrial fluorescence was mostly red in LANCL1/2-overexpressing cells, while it was predominantly green in cells silenced for both LANCL proteins. The calculated red/green ratio was approx. one log higher in LANCL1/2-overexpressing compared with double silenced cells, reflecting a steeper ∆Ψ (Figure 1E, right panel, white bars). The red/green ratios of H9c2 cells transformed with the two different control vectors (PLV and SCR) were similar, and they were between those of the overexpressing and double-silenced cells (Appendix A, panel B). This mitochondrial ∆Ψ difference was not due to the virus vectors, but to overexpression or silencing of the LANCL1/2 proteins. Cell treatment with cyclosporin A (CsA), an inhibitor of both types of proton “leakers”, ATP synthase [19] and the ATP/ADP translocator ANT-1 [20], further significantly increased (by 37%) the already high ∆Ψ in overexpressing cells (Figure 1E); the lower percentage increase in the red/green fluorescence ratio in overexpressing vs. double-silenced cells (37 vs. 62%) may indicate the presence of other, CsA-insensitive, proton “leakers” in LANCL1/2-overexpressing cardiomyocytes.

### 3.2. LANCL1/2-Silencing Reduces, and Their Overexpression Increases, Oxidative Metabolism Gene Expression, Glucose Uptake and NAD/ATP Content

The higher respiration rates, both basal and maximal, observed in LANCL1/2-overexpressing vs. -silenced H9c2 need to be sustained by a higher oxidative metabolic rate. Indeed, the transcription of glucose transporters GLUT4 and GLUT1, of glycolytic enzymes (phosphofructokinase-1, PFK1, glyceraldehyde dehydrogenase, GAPDH, pyruvate kinase, PK), of subunit one of pyruvate dehydrogenase (PDHα1, required for pyruvate entry into the Krebs cycle), of proteins involved in fatty acid transport (carnitine palmitoyltransferase, CPT1β) and oxidation (acyl-coenzyme A dehydrogenase, ACADS), and of fibroblast growth factor 21 (FGF21), a hormone regulating energy metabolism at a tissue and organismic level, all increased approx. two-fold in LANCL1/2-overexpressing cells compared with their controls, infected with the empty vector PLV and treatment with 100 nM ABA further increased mRNA levels. The same mRNAs were conversely significantly reduced in double-silenced cells compared with their respective controls, transfected with the scrambled sequences (SCR) used for LANCL1/2 silencing (Figure 2A, upper panel).

In addition to these metabolism-controlling genes, transcription of other genes involved in mitochondrial function was also upregulated in LANCL1/2-overexpressing and conversely downregulated in double-silenced H9c2: subunit 1 of complex I of the respiratory chain (MT-ND1), uncoupling proteins 1 and 3 (UCP1, UCP3) and the adenine nucleotide translocator ANT1, which is necessary to allow ADP/ATP exchange across the inner mitochondrial membrane, but also mediates fatty acid transport that partly dissipates the proton gradient [20], similarly to UCP3 (Figure 2A, lower panel).

mRNA levels of NAD-synthesizing NAMPT and of NO-producing eNOS were also significantly upregulated in overexpressing compared with double-silenced cells, approx. 25-fold, as already observed [10], and control PLV- and SCR-infected cells showed mRNA levels between those of double-overexpressing and double-silenced cells (Appendix A, panel C). The logarithmic increase in the transcription levels of ANT1, UCP1 and UCP3 in LANCL1/2-overexpressing vs. double-silenced H9c2 was in agreement with the higher proton leak observed in these cells compared with the double-silenced cells (Figure 1D). The mitochondrial-to-genomic DNA ratio increased almost three times in LANCL1/2-overexpressing cells compared with their controls (PLV) while it was conversely severely reduced in double-silenced cells; thus, overexpressing cells showed a 12-fold higher mitochondrial/genomic DNA ratio compared with double silenced cells; the fact that mitochondrial fluorescence increased 3-fold while the mitochondrial-to-genomic DNA ratio increased 12-fold in overexpressing vs. double silenced cells may reflect not only an increase in mitochondrial number, but also a higher mitochondrial DNA content. Finally, treatment with 100 nM ABA of the overexpressing, but not of the double-silenced cells, further significantly increased the mitochondrial/nuclear DNA ratio, which was approx. 25-times higher in overexpressing vs. double-silenced cells (Figure 2A, lower panel).

In line with the increased transcription of glucose transporters GLUT1 and GLUT4, glucose uptake, as measured with the fluorescent glucose analog 2-NBDG, was higher in LANCL1/2-overexpressing than in double-silenced cells (by approx. 40%) and, further, was significantly increased to three-times higher values by pre-incubation with 100 nM ABA in the overexpressing, but not in the silenced cells (Figure 2B, left panel). An increased energy production in LANCL1/2-overexpressing vs. silenced H9c2 was also evident from their respective ATP/ADP ratio, which was approx. five vs. three, and their NAD^+^ content (Figure 2B, central and right panels).

Finally, mRNA levels of three different ion channels critical for cardiomyocyte electrical conductance and contractile activity were investigated; the two-pore domain potassium channel TREK-1, the voltage-gated sodium channel SCN1B and the L-type calcium channel CACNA1c. mRNAs specific for SCN1B and CACNA1c increased approx. eight-fold in LANCL1/2-overexpressing vs. control cells, and treatment with ABA further increased transcription. In contrast, mRNA levels for TREK-1 were not significantly higher and increased only slightly upon ABA treatment in overexpressing cells. Transcription of SCN1B and CACNA1c was conversely significantly reduced in double-silenced cells compared with their controls (SCR) and again transcription of TREK-1 was not significantly affected by LANCL1/2 silencing (Figure 2C).

### 3.3. Cell Volume and Proliferation Rate Increase in LANCL1/2-Overexpressing Compared with Double-Silenced Cells

As observed under a phase contrast microscope during routine cell culture maintenance, LANCL1/2-overexpressing cells appeared visibly larger and more rapidly dividing than double silenced cells. To quantify these visual impressions, cell volume, protein content and doubling time were analyzed.

The cell volume was measured by confocal microscopy on calcein-loaded cells (Figure 3A, left panel). While cell volume was not significantly different between PLV and SCR cells, LANCL1/2-overexpressing cells were significantly larger (175%) than double-silenced cells; compared with their respective controls, LANCL1/2-overexpressing cells were 45% larger than PLV-infected cells and double-silenced cells were 30% smaller than SCR controls (Figure 3A, upper right panel).

As could be anticipated from the larger cell volume, total cell protein content per 10^6^ cells was also higher in LANCL1/2-overexpressing compared with double-silenced cells (56% higher) and again control cells (SCR and PLV) had a similar protein content (Figure 3A, lower right panel), indicating that the difference in cell volume and in protein content between LANCL1/2-overexpressing and double-silenced cells was not attributable to the distinct vectors used for their transformation.

To evaluate cell doubling time in LANCL1/2-overexpressing vs. double-silenced cells, the time needed to double cell culture proteins relative to time = zero values was compared (Figure 3B, upper left panel). Cell doubling time was similar in control cells (SCR and PLV, approx. 70 h), conversely, it was significantly reduced in LANCL1/2-overexpressing cells (35 h) and, conversely, increased in double-silenced cells (120 h) (Figure 3B, upper right panel).

The reduced doubling time for the LANCL1/2-overexpressing cells compared with double-silenced cells was also visually evident from photomicrographs taken during cell culture (Figure 3B, lower right panels), while control cells (SCR and PLV) showed a similar cell density during culture (Figure 3B, lower left panels).

The four-fold higher growth rate in LANCL1/2-overexpressing vs. double-silenced cells prompted us to analyze the mRNA levels of a selection of cyclins (CCNs) and of cyclin-dependent kinases (CDKs) whose overexpression or targeted delivery increases or induces cardiomyocyte proliferation in vitro and in vivo [21,22,23,24,25]. In fact, LANCL1/2-overexpressing cells showed a strongly increased transcription of all the CCNs and CDKs explored compared with their controls; those transfected with PLV, and treated with 100 nM ABA further upregulated mRNA levels (Figure 3C). Conversely, double silenced cells had a markedly reduced transcription of the same genes, compared with their controls; transfection with scrambled sequences (SCR) and ABA treatment did not induce any increase in gene transcription (Figure 3C). In particular, CCNA2, CCND1 and CDK4 induce cell proliferation when overexpressed in adult cardiomyocytes [24,25].

Altogether, these results indicate that LANCL1/2-overexpressing cardiomyocytes have an increased protein content, are larger and grow faster than double-silenced cells. All these features are likely supported by the increased energetic proficiency of their mitochondria.

### 3.4. Quantitative and Qualitative Differences in the Expression of Contractile and Cytoskeletal Proteins in LANCL1/2-Overexpressing vs. Double-Silenced H9c2 Cells

The different cell volumes and protein contents observed in the double-silenced vs. overexpressing cells suggested we compare the cytoskeletal and contractile proteins of the two cell types. Thus, we analyzed transcription and expression of the following functional proteins of cardiomyocytes: cytoskeletal tubulin (TUBB2A), the contractile complex of actin (ACTC1) and heart-specific myosin heavy chain 7 (MYH7), proteins involved in cell-to-cell contact connexin-43 (Cx43) and in cytoskeletal stabilization (β-catenin, CTNNB1). The transcription of the contractile complex genes ACTC1 and MYH7 increased by approx. 40% in OVL1+2 cells compared with controls (PLV) and further increased by approx. 150% in the presence of 100 nM ABA; transcription of these genes was conversely significantly reduced in SHL1+2 compared with their controls (SCR) and ABA treatment of the cells did not induce significant modifications in their transcription (Figure 4A).

mRNA levels of Cx43 and β-catenin increased slightly in overexpressing cells compared with their controls (by approx. 30%) and were instead significantly reduced (by approx. 50%) in double-silenced cells compared with their respective controls (Figure 4A).

To confirm the different expression levels of the cytoskeletal and contractile proteins, an immunofluorescence test was performed by confocal microscopy [26]. As shown in Figure 4B, LANCL1/2-overexpressing cells had more F-actin fibers than double-silenced cells; in the latter cells, the filaments also seemed to be less organized. In addition, overexpressing cells also showed increased fluorescence for α-tubulin and MYH7 when compared with double-silenced cells. The cell distribution of β-catenin also appeared to be different in overexpressing vs. double-silenced H9c2; in the latter cells, staining for β-catenin appeared to be more punctate, perhaps as a consequence of some rearrangement of F-actin fibers, which interact with β-catenin at the membrane level; Huber et al. 2001 [27]. Moreover, the distribution of Cx43, whose trafficking is at least in part regulated by microtubules, Giepmans 2001 [28], was also different in LANCL1/2-overexpressing vs. double-silenced cells; in LANCL1/2-overexpressing cells, it showed a diffuse staining all over the cytosol, whereas in LANCL1/2-silenced cells, it acquired a marked perinuclear localization, suggesting retention of the protein in the Golgi and less protein available for gap junction formation at the plasma membrane.

### 3.5. ERRα Mediates the Transcriptional Effects Induced by LANCL1/2-Overexpression in H9c2

We previously observed that overexpression of LANCL1/2 in human adipocytes differentiated from immortalized preadipocytes induced a 10- and 5-fold increase in ERRα transcription in brown and beige adipocytes, respectively, compared with similarly differentiated control cells not overexpressing LANCL proteins [10]. In addition, together with PGC-1α, ERRα controls the transcription of several genes involved in mitochondrial-energy production in cardiac and skeletal muscle [7] and may thus be involved in some of the mitochondrial activities stimulated in LANCL1/2-overexpressing H9c2.

Firstly, we confirmed in H9c2 that overexpression of the LANCL proteins induces an increased transcription of ERRα (four-fold) compared with PLV-infected control cells, and that double silencing of the LANCL proteins instead results in a significantly reduced transcription (by approx. 80%) of ERRα, compared with scrambled-transfected control cells. The combination of these opposite transcriptional trends leads to approx. 20-fold higher mRNA levels of ERRα in LANCL1/2-overexpressing vs. double-silenced cells (Figure 5A, left panel). The silencing of ERRα in untransformed H9c2 cardiomyocytes (SHERRα, not overexpressing LANCL1/2) allowed us to investigate whether there was a reciprocal transcriptional control between ERRα and the LANCL proteins. Indeed, silencing of ERRα significantly reduced (by approx. 75%) the transcription of endogenous (not overexpressed) LANCL1 and LANCL2 and also abrogated the stimulatory effect of ABA on LANCL1/2 transcription (Figure 5A, right panel). Thus, a reciprocal transcriptional activation links ERRα and the LANCL proteins.

As shown in Figure 5B, the knockdown of ERRα in LANCL1/2-overexpressing cells (OVL1+2-SHERRα), where ERRα mRNA levels are spontaneously four-times higher than in PLV-infected cells, as a consequence of the LANCL protein overexpression (Figure 5A), was tested with both immunoblot (Figure 5B, left and central panels) and qPCR (Figure 5B, right panel). A similar, approx. 75%, reduction was observed both in protein expression and in mRNA transcription relative to OVL1+2-SCR (Figure 5B).

We next investigated the effect of ERRα silencing on the transcription of the genes that we previously identified as part of the signaling pathway downstream of the ABA/LANCL system in muscle cells, i.e., the AMPK/PGC1α/Sirt1 axis [9]. The transcription of these mRNAs was significantly reduced (by approx. 90%) in LANCL1/2-overexpressing cells silenced for ERRα (OVL1+2-SHERRα) relative to controls, transfected with scrambled sequences (OVL1+2-SCR) (Figure 5C, upper panels).

All other target genes previously shown to be transcriptionally upregulated in LANCL1/2-overexpressing H9c2 (Figure 2A), and lying downstream of the AMPK/PGC-1α/Sirt1 axis, showed a similar, significant (>80%) reduction in their mRNA levels in ERRα-silenced, LANCL1/2-overexpressing H9c2 compared with control cells (OVL1+2-SCR) (Figure 5C, upper panels). In addition, ERRα silencing in LANCL1/2-overexpressing cells caused a very severe reduction in the transcription of those genes relevant to mitochondrial function (NAMPT, eNOS, MT-ND1, ANT1 and UCP1, Figure 5C, upper right panel), which were upregulated in LANCL1/2-overexpressing compared with double-silenced cells (Figure 2A). ERRα proved also essential in mediating the stimulation induced by LANCL1/2 overexpression, and further increased by ABA treatment of the cells, of the transcription of ion channels (Figure 5C, left central panel), of cytoskeletal and contractile proteins (Figure 5C, right central panel) and of cell cycle-controlling cyclins and CDKs (Figure 5C, lower panel). The amplitude of the transcriptional inhibition elicited by its silencing in LANCL1/2-overexpressing H9c2 clearly identifies ERRα as the major transcription factor orchestrating the multifaceted-transcriptional regulation exerted by the ABA/LANCL system in cardiomyocytes.

### 3.6. ERRα Mediates the Functional Effects Induced by LANCL1/2-Overexpression in H9c2

To explore the functional consequences of ERRα silencing on LANCL1/2-overexpressing H9c2 we next investigated some of the effects downstream of the regulated genes, i.e., proton gradient amplitude, cell volume and cell proliferation.

ERRα silencing greatly reduced the proton gradient of native H9c2 (Appendix A) and even more evidently in LANCL1/2-overexpressing cells (OVL1+2-SHERRα), as shown by the predominantly green mitochondrial fluorescence and by the greatly reduced red/green ratio of JC-1-loaded cells compared with LANCL1/2-overexpressing cells transfected with scrambled sequences (SCR) (Figure 6A). The approx. 25-fold reduction in the red/green fluorescence ratio in ERRα-silenced vs. control LANCL1/2-overexpressing H9c2 indicates that ERRα regulates the proton gradient formation in LANCL1/2-overexpressing H9c2 cardiomyocytes. Indeed, the reduced transcription of subunit 1 of respiratory complex I (MT-ND1) and of the ADP/ATP translocator ANT1 observed in ERRα-silenced cells (Figure 5C) is expected to negatively affect mitochondrial ∆Ψ. 

Cell volume was also reduced in ERRα-silenced, LANCL1/2-overexpressing cells compared with controls, as measured using calcein-AM; ERRα silencing reduced cell volume by approx. 60% (Figure 6B).

Finally, cell-doubling time was significantly increased in ERRα-silenced, LANCL1/2-overexpressing cells compared with overexpressing cells transfected with the empty vector (controls), as inferred from the time needed to double the total protein content of cell cultures (Figure 6C).

We previously observed a causal role for the increased NO generation by LANCL1/2-overexpressing cells in mediating some of the beneficial effects observed in these cells after hypoxia/reoxygenation [5]. Interestingly, the transcription of ERRα also appears to be NO-dependent: in the presence of the NOS inhibitor L-NAME, mRNA levels for ERRα were significantly reduced (by approx. 70%) in LANCL1/2-overexpressing H9c2 (Figure 5A, left panel). Thus, ERRα controls transcription of eNOS (Figure 5C) and NO in turn controls ERRα transcription, generating a feed-forward mechanism which may explain the high levels of NO production measured in LANCL1/2-overexpressing H9c2 [5].

## 4. Discussion

Taken together, the results reported in this study allow us to conclude that the overexpression of LANCL1/2 significantly improves, while their combined silencing dramatically reduces, several key functional features of rat H9c2 cardiomyocytes (Figure 7).

As compared with double-silenced cells, LANCL1/2-overexpressing H9c2 cells show: (i) increased mitochondrial respiration, with higher basal and maximal respiration rates, a doubling of the spare respiratory capacity and a steeper proton gradient (ΔΨ) (Figure 1D,E); (ii) increased fatty acid-fueled respiration rate (Figure 1D); (iii) increased NO generation [5]; (iv) increased transcription and expression of contractile and ion channel proteins (Figure 4); (v) improved resistance to hypoxia/reoxygenation [10]; (vi) increased proliferation rates (Figure 3). In short, LANCL1/2 overexpression, and their targeted stimulation by treatment of the cells with ABA, transforms H9c2 cardiomyoblasts into “super-cells”. How are such pleiotropic effects orchestrated?

The AMPK/PGC-1α/Sirt1 axis is known to control energy metabolism and mitochondrial respiration in skeletal muscle [29,30], adipose tissue [31,32] and the heart [33,34,35]; thus, the fact that LANCL1/2 overexpression induces the activation of this signaling axis (via transcriptional and post-transcriptional mechanisms) in adipocytes [10], skeletal muscle cells [9] and cardiomyocytes [5] is expected to underlie most of the functional effects observed in overexpressing H9c2. A significant new piece of information added in this study is that the transcription factor ERRα is directly responsible for all transcriptional effects observed in LANCL1/2-overexpressing cells and is functionally linked to the LANCL proteins, with a reciprocal feed-forward mechanism of transcriptional stimulation (Figure 7).

ERRα activity is known to be important for cardiomyocyte mitochondrial function. The critical role of ERRα in cardiomyocyte function starts during myocyte maturation [36] and continues in adult cardiomyocytes, in combination with PGC-1α, with which it shares a reciprocal transcriptional positive regulation [6,37]. Thus, ERRα/PGC-1α control cardiac energy metabolism, metabolic flexibility, mitochondrial respiration and biogenesis. Indeed, pharmacologic targeting of AMPK, which activates ERRα transcription and promoter activity [38] and/or of the ERRα/PGC-1α system [39,40] has been proposed to improve cardiac function in the ailing heart.

Here, we show that LANCL1/2 overexpression increases, and their double silencing conversely significantly reduces ERRα transcription and expression. In turn, silencing of ERRα reduces endogenous LANCL1/2 mRNA levels in H9c2 and significantly reduces or abrogates all transcriptional and functional effects induced by LANCL1/2-overexpression (Figure 5 and Figure 6). It may be surprising that silencing just one transcription factor (ERRα) should have such wide-ranging effects on cardiomyocyte physiology; however, one should consider that all transcriptional and functional effects studied here are closely related; an increased oxidative metabolism is needed to allow higher rates of ATP production, in turn allowing increased protein synthesis and accelerated cell proliferation. Indeed, cyclin levels control transcription of several metabolism-related genes, whose enzyme products are necessary for energy production to allow cell duplication [41].

A role for ERRα in the control of cell proliferation has been previously described in lung cancer cells [42]. However, to our knowledge, the results reported here (Figure 5C) are the first direct evidence that silencing of ERRα significantly reduces the transcriptional levels of several cyclins and negatively affects cardiomyocyte proliferation (Figure 6).

Although adult cardiomyocytes have lost their proliferative potential, specific cyclin (CCN)/cyclin-dependent kinase (CDK) complexes have been shown to be able to re-start proliferation when overexpressed in adult cardiomyocytes: CDK4/CCND and CDK2/CCND complexes promote entry into G1-S phase [25]; overexpression of CCNA2 induces a proliferative response in adult porcine cardiomyocytes in vivo and in vitro [24]; the targeted expression of cyclin D2 induces cardiomyocyte proliferation and infarct regression in mice [22]. These cell cycle-controlling genes are among the ones whose transcription increases in LANCL1/2-overexpression, and decreases in double-silenced H9c2 (Figure 3C), indicating a transcriptional control by the LANCL proteins on these cell cycle regulators.

Cell cycle-controlling cyclins, kinases and transcription factors, in particular, CCNDs and E2Fs, besides being essential for cell cycle progression, also play important roles in the regulation of energy metabolism [41]: indeed, nutrient availability, metabolic energy production, mitochondrial activity and cell division are tightly linked processes, which need a coordinated regulation. The fact that the overexpression of LANCL1/2 significantly increases the transcription of several cell cycle- and metabolism-controlling cyclins via ERRα, and that, conversely, their combined silencing dramatically reduces mRNA levels, identifies the ABA/LANCL1-2/ERRα system as a new regulator of this complex gene system, warranting further studies to deepen our understanding of the role of this hitherto unknown relationship in pathological heart conditions, which may benefit from its pharmacologic stimulation.

Increased mitochondrial respiratory activity is generally considered conducive to increased ROS production, as all respiratory complexes (RC) can produce ROS when electron flow is excessive, causing an electron “overflow” at the reduced RC, or hampered by a limitation of the terminal electron acceptor, O_2_.

Mild proton leak across the inner mitochondrial membrane has been recently acknowledged as a means to improve respiratory chain function, while at the same time reducing ROS generation [43]. The general consensus is that, by reducing the ΔG for proton pumping, a mild proton leak facilitates electron transport and reduces ROS generation at the RC, at the expense of some ATP. However, we observed both a higher proton leak and an increased ATP-dependent respiration in LANCL1/2-overexpressing vs. double silenced cells (Figure 1D), indicating that the “fine-tuning” of the respiratory chain works better in overexpressing vs. silenced cells, allowing a higher respiration rate and ATP-production and a reduction in mitochondrial-ROS content in the face of an increased ΔΨ. Proton transport associated with fatty acid (FA) translocation by ANT1 and with the activity of ATP-synthase have been identified as mechanisms responsible for partial dissipation of the ΔΨ [19,20].

ANT1, whose main function is to translocate ATP and ADP across the inner mitochondrial membrane, also allows an FA-dependent “proton leak” through the inner mitochondrial membrane [20,44]. ANT1 and ATP synthase are both inhibited by CsA [19] and indeed a quantitatively marked increase in the ΔΨ occurs in CsA-treated LANCL1/2-overexpressing cells, further increasing their already high gradient (Figure 1E). This conspicuous CsA-sensitive proton leak in LANCL1/2-overexpressing cells (Figure 1E) may play a role in “easing” the work of the proton pumps of the respiratory chain. Indeed, overexpressing cells show a significantly higher FA-dependent percentage of proton leak and of maximal OCR than double-silenced cells, while the percentage of FA-dependent ATP-linked OCR is similar in the two cell types (Figure 1D), suggesting that the primary function of FA mitochondrial utilization in overexpressing cells is not to produce ATP. Apart from ANT1, the transcription of UCP1 and UCP3 is also significantly increased in overexpressing compared with double-silenced cells (approx. seven and eight-times higher), likely indicating a role also for these proton leakers in the higher OXPHOS efficiency in overexpressing vs. silenced cells.

The mechanisms allowing an unconstrained and regular flow of electrons from NADH (and FADH_2_) to oxygen as the terminal electron acceptor (TEA) are also important to prevent ROS production during electron transfer at the RC, since they can all become sites of ROS production through retrograde electron transport, typically under limited oxygen availability [45,46,47]. Indeed, ROS production at RC I, II and III is believed to occur both physiologically and under pathological conditions and ROS are considered important regulatory signals of cell energy metabolism and respiration [48,49,50,51]. Thus, it is reasonable to assume that an increased oxidative metabolic rate may “overflow” the RC with electrons, exceeding their ability for electron transport and resulting in ROS generation. A mechanism “easing” electron flow through the RC is the availability of other TEAs that can prevent the retrograde electron transport from reduced respiratory complexes, limiting ROS production, while at the same time allowing coenzyme re-oxidation. Fumarate reduction to succinate has long been known as a mechanism capable of alleviating electron “overflow” at the succinate dehydrogenase-coenzyme Q site of the respiratory chain [52] and fumarate has recently been rediscovered as a TEA and an important means to allow complex I proton pumping under conditions of limited O_2_ availability [53]. By providing an electron “leak” from the respiratory chain, fumarate also prevents ROS production at the RCs while at the same time allowing some proton pumping to occur via complex I. Whether this mechanism of electron leak is more active in LANCL1/2-overexpressing vs. double-silenced cells remains to be investigated.

The ROS-generating reverse flow of electrons at the RC, which can physiologically occur under changing conditions of oxygen and substrate availability, is particularly sensitive to the mitochondrial membrane potential. Heinen et al. found that under substrate conditions that allow reverse electron flow at complex II, K^+^ influx into the matrix through Ca^2+^-sensitive K^+^ channels (thus, depolarizing the intermembrane space) accelerated the forward electron flow and concomitantly reduced ROS production in guinea pig heart mitochondria [54]. Ross et al. observed that a slight depolarization of ΔΨ reduces reverse electron flow-induced ROS generation at complex I in murine heart mitochondria, indicating that reverse electron flow at complex I is particularly sensitive to depolarization [55]. Miwa and Brand found that mild uncoupling significantly reduced ROS production from complex I on the matrix side of the membrane of insect mitochondria [56]. Thus, several observations support the notion that a high ΔΨ is favorable for ROS generation, while mild proton gradient dissipation reduces ROS generation.

To sum up, both a proton “leak” from the intermembrane space and an electron “leak” from the respiratory chain flow are mechanisms of OXPHOS fine tuning, which can be activated in mitochondria to adapt to quantitative changes in the flow of charges in order to maximize energy production and limit ROS generation. The ABA/LANCL system, via ERRα, appears to take advantage of, or to regulate, some of these OXPHOS fine-tuning mechanisms. Indeed, despite a two-fold increase in the proton leak in overexpressing cells, the spare respiratory capacity of these cells is approximately twice that of double-silenced cells (Figure 1D).

The term “mitohormesis” is used to describe a complex and, so far, incompletely understood mitochondrial response to “stress”, such as oxygen deprivation, excess ROS production or nutrient deficiency [57]. Interestingly, several of the transcriptional and functional effects observed in LANCL1/2-overexpressing H9c2 fall within this concerted response. Mild OXPHOS uncoupling via increased transcription of several H^+^ transporters (UCP1, UCP3, ANT1, Figure 2A), leading to an increased proton leak (Figure 1D), increased mitokine production (FGF21, Figure 2A), exerting mitochondrial and nuclear autocrine and paracrine effects, are among the keynote features of mitohormesis. This multifaceted response is orchestrated through the activation of AMPK, PGC-1α and Sirt1, which is indeed the main signaling axis activated by the ABA/LANCL system, not only in H9c2 cardiomyocytes [5], but also in human brown adipocytes, where a similar increase in mitochondrial respiration rate and ΔΨ were observed in overexpressing cells [10]. The fact that several keynote mitochondrial responses typical of mitohormesis are among those activated by the ABA/LANCL system allows us to hypothesize that this hormone/receptor system controls mitohormesis, whose function may not be limited to the response to mitochondrial “stress” conditions, such as hypoxia, but could more generally serve the purpose of optimizing mitochondrial function under conditions of changing oxygen availability, electron-flow intensity and ATP requirements. Indeed, some of these mitochondrial adaptations to increased oxidative metabolism occur in the skeletal muscle and in the heart under conditions of physical exercise, arguably a “physiological” condition.

Several questions arise from the results described in this and in the previous reports [5].

Can we use ABA to activate the LANCL1/2-ERRα signaling pathway and improve cardiomyocyte function or resilience to stress conditions? Targeting the PGC-1α/Sirt1/ERRα axis has been proposed as a means to improve cardiomyocyte function in diabetic cardiomyopathy [39]; the identification of the ABA/LANCL1-2 system as an essential part of the ERRα activating pathway may provide a new pharmacological agonist (ABA) and new molecular targets (the LANCL proteins) to this end.

Another open question regards the molecular signal that activates endogenous LANCL1/2 transcription in cardiomyocytes. NO may indeed be the first signal initiating the functional responses downstream of the LANCL1/2-ERRα signaling pathway, as it is normally produced by the beating heart and cardiac NO levels are affected by conditions of cardiomyocyte “stress” [58]. NO is apparently involved in a positive feedback mechanism linking eNOS transcription and activity to the expression levels of LANCL1/2 and ERRα. Thus, once activated, a reciprocal feed-forward mechanism maintains a transcriptional activation of LANCL1/2 and ERRα and NO generation in cardiomyocytes (Appendix A).

It is noteworthy that LANCL1/2 overexpression and ABA-treatment in H9c2 also significantly increase transcription of GPTCH [5], the rate-limiting enzyme in the synthesis of TBH4, the coenzyme needed to prevent “uncoupling” of NOS, resulting in ROS instead of NO generation [59]. In addition, TBH4 also stimulates mitochondrial biogenesis and cardiac contractility via PGC-1α [60], possibly participating in the LANCL-ERRα-coordinated improvement of mitochondrial performance.

## 5. Conclusions

As inferred from the experimental results obtained in rat H9c2 cardiomyoblasts, the ABA/LANCL1-2 hormone/receptor system emerges as a new controller of cardiomyocyte “fitness” via a reciprocal transcriptional stimulation with ERRα. Downstream of this signaling axis, mitochondrial biogenesis and respiration, energy metabolism, cytoskeletal and contractile protein synthesis and cell cycle are accelerated. It is likely that several hormonal/stress signals (including ABA and NO) activate this signaling pathway under physiological conditions, and their action may be exploited to improve organ function under pathological conditions. The results described here need to be confirmed on primary culture rodent cardiomyocytes and/or on iPSC-derived human cardiomyocytes before attempting in vivo translation of these promising results.

## Figures and Tables

**Figure 1 antioxidants-12-01692-f001:**
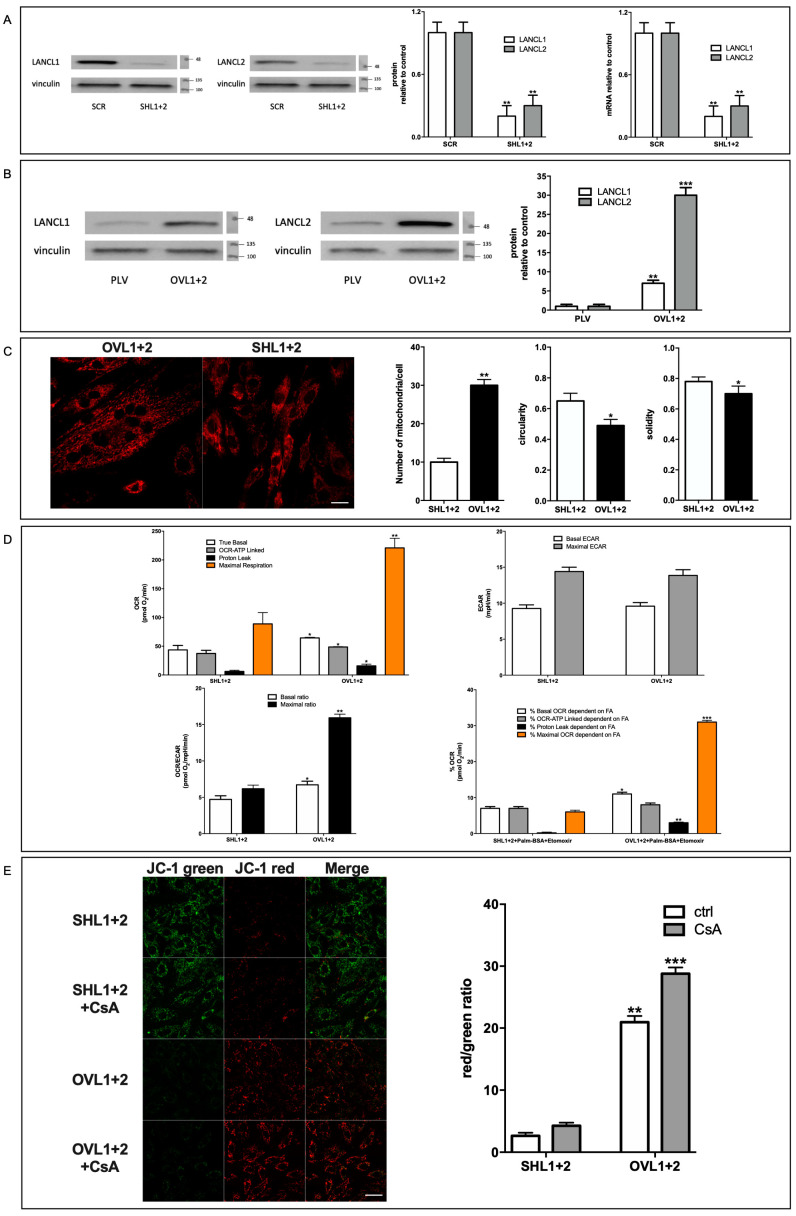
Mitochondrial proton gradient and respiration in H9c2 cardiomyocytes are controlled by the expression level of LANCL1/2. LANCL1 and LANCL2 proteins were stably silenced (**A**) or overexpressed (**B**) in H9c2 rat cardiomyocytes by viral infection. (**A**) Left panels, representative Western blots of LANCL1/2 in double-silenced cells (SHL1+2); central panel, densitometric quantitation of the LANCL proteins relative to control cells, transfected with the vector containing scrambled silencing sequences (SCR); right panel, LANCL1/2 mRNA levels in LANCL1/2-silenced cells relative to SCR. Values are normalized on vinculin. ** *p* < 0.01 relative to SCR control cells by unpaired *t*-test. Data shown are the mean ± SD of 3 experiments per group, with each value calculated in triplicate. (**B**) Left panels, representative Western blots of LANCL1 and LANCL2 proteins in cells overexpressing both proteins (OVL1+2); right panel, densitometric quantitation of the LANCL proteins relative to control cells, transfected with the empty vector (PLV). Values are normalized on vinculin. ** *p* < 0.01 and *** *p* < 0.005 relative to PLV control cells by unpaired *t*-test. Data shown are the mean ± SD of 3 experiments per group, with each value calculated in triplicate. The mitochondrial number was evaluated by MitoTracker analysis in OVL1+2 and SHL1+2 cells. (**C**) Left panels, representative confocal microscopy images of overexpressing and silenced cardiomyocytes; the mean number of mitochondria per cell was calculated by counting mitochondria in approx. 50 cells per type; right panels, morphological parameters of circularity and solidity in the same cells. The mean ± SD of the relative mitochondrial fluorescence was always calculated in at least 4 microscopic fields (scale bar: 20 µm). * *p* < 0.05 and ** *p* < 0.01 relative to SHL1+2 cells by unpaired *t*-test. Respiration measurements were performed using the Seahorse XFp analyzer, with the sequential addition of oligomycin, FCCP and rotenone/antimycin A. (**D**) Oxygen consumption rates (OCR, upper left panel), extracellular acidification rate (ECAR, upper right panel), OCR/ECAR ratio (lower left panel) and the percentage of the OCR dependent on fatty acid oxidation (lower right panel) were measured in SHL1+2 and OVL1+2 cells. * *p* < 0.05, ** *p* < 0.01 and *** *p* < 0.005 relative to SHL1+2 cells (untreated for upper panels and lower left panel or treated with Palm-BSA + Etomoxir for lower right panel) by unpaired *t*-test. Data shown are the mean ± SD of 4 experiments per group, with each value calculated in triplicate. Cardiomyocytes were loaded with the ∆Ψ-sensitive ratiometric fluorescent dye JC-1 and cultured with or without 4 µM CsA for 2 h. (**E**) Left panel, representative confocal microscopy images; right panel, red/green fluorescence ratio calculated in at least 4 microscopic fields (scale bar: 20 µm) for each experiment. ** *p* < 0.01 and *** *p* < 0.005 relative to SHL1+2 cells by unpaired *t*-test.

**Figure 2 antioxidants-12-01692-f002:**
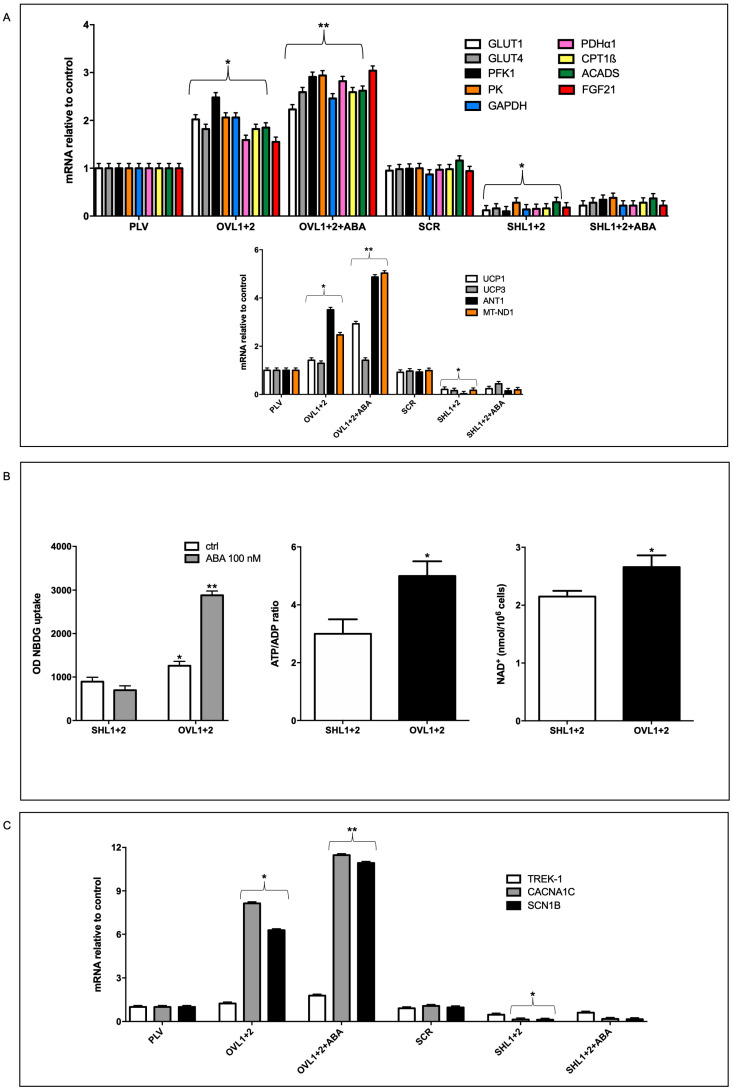
LANCL1/2-overexpression increases, and their double silencing reduces mRNA levels of oxidative metabolism genes, glucose uptake and NAD/ATP content. (**A**) Transcription of the indicated genes in cells overexpressing or silenced for the expression of LANCL1 and LANCL2 proteins by qPCR analysis and incubated in the absence or in the presence of 100 nM ABA for 4 h. Data are expressed relative to control cells (PLV or SCR). Upper panel, GLUT1, GLUT4, PFK1, PK, GAPDH, PDHα1, CPT1β, ACADS and FGF21 mRNA levels; lower panel, UCP1, UCP3, ANT1 and MT_ND1 mRNA levels. * *p* < 0.05 relative to the respective untreated control cells (PLV or SCR) and ** *p* < 0.01 relative to OVL1+2 cells by unpaired *t*-test. Results shown are the mean ± SD of 3 experiments per group. (**B**) Glucose uptake (left panel), ATP/ADP ratio (central panel) and NAD^+^ content (right panel) were measured in the same cells. * *p* < 0.05 and ** *p* < 0.01 relative to SHL1+2 cells by unpaired *t*-test. Data shown are the mean ± SD of 4 experiments per group. (**C**) qPCR analysis of the transcription of TREK-1, CACNA1C and SCN1B in cells overexpressing or silenced for the expression of LANCL1 and LANCL2 proteins and treated with or without 100 nM ABA for 4 h. Data are expressed relative to control cells (PLV or SCR). * *p* < 0.05 relative to the respective untreated control cells (PLV or SCR) and ** *p* < 0.01 relative to OVL1+2 cells by unpaired *t*-test. Data shown are the mean ± SD of 3 experiments per group.

**Figure 3 antioxidants-12-01692-f003:**
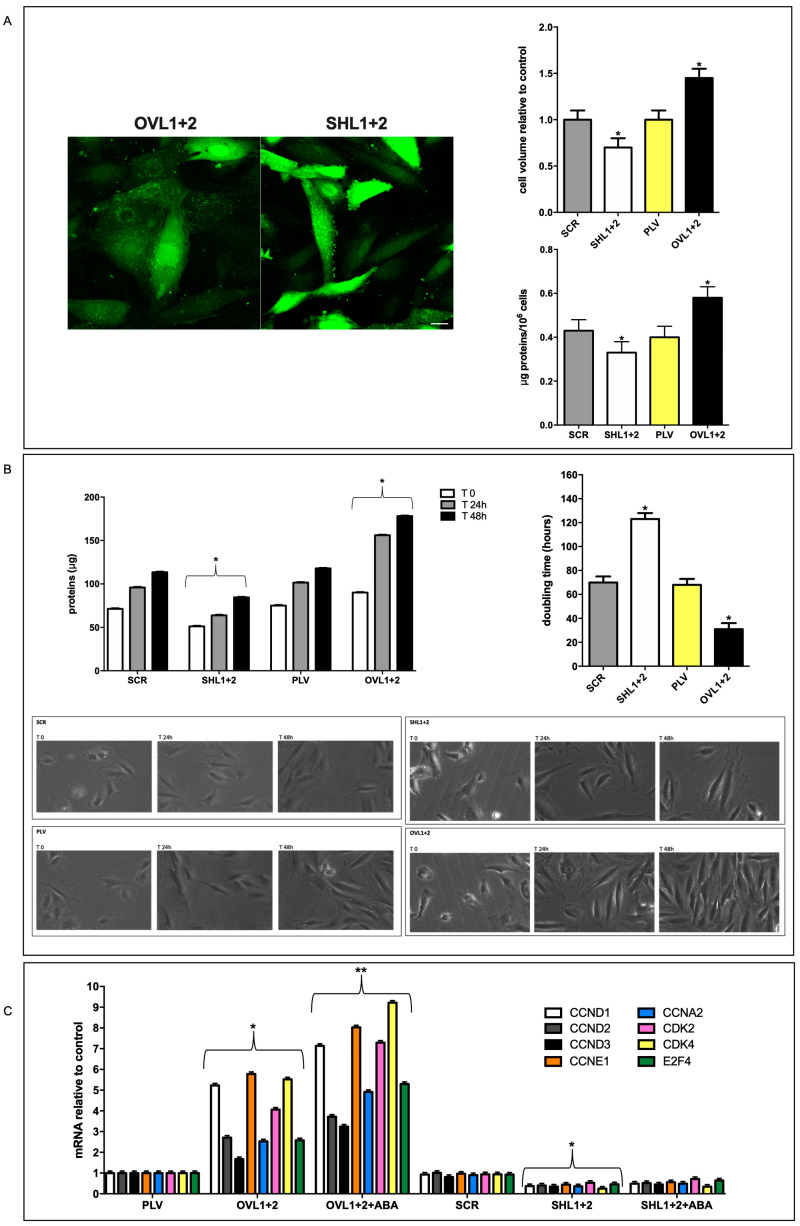
Effect of LANCL1/2 overexpression or silencing on cardiomyocyte volume and proliferation. (**A**) Cell volume by confocal microscopy on calcein-loaded cells and total cell protein content per 10^6^ cells were measured. Left panel, representative confocal microscopy images; upper right panel, densitometric quantitation relative to the respective untreated control cells (PLV or SCR) calculated in at least 3 microscopic fields (scale bar: 20 µm) for each experiment; lower right panel, cell protein content reported as µg proteins/10^6^ cells. * *p* < 0.05 relative to the respective untreated control cells (PLV or SCR) by unpaired *t*-test. Results shown are the mean ± SD of 4 experiments per group. (**B**) Cell protein content after 24 and 48 h with respect to time = zero (upper left panel) and doubling time (upper right panel) were calculated. Representative photomicrographs were taken during the cell culture (lower panels, 20× magnification). * *p* < 0.05 relative to the respective untreated control cells (PLV or SCR) by unpaired *t*-test. Results shown are the mean ± SD of 4 experiments per group, with each value calculated in triplicate. (**C**) qPCR analysis of the transcription of a selection of cyclins (CCNs) and of cyclin-dependent kinases (CDKs) in OVL1+2 or SHL1+2 cells and incubated with or without 100 nM ABA for 4 h. Data are expressed relative to control cells (PLV or SCR). * *p* < 0.05 relative to the respective untreated control cells (PLV or SCR) and ** *p* < 0.01 relative to OVL1+2 cells by unpaired *t*-test. Results shown are the mean ± SD of 4 experiments per group.

**Figure 4 antioxidants-12-01692-f004:**
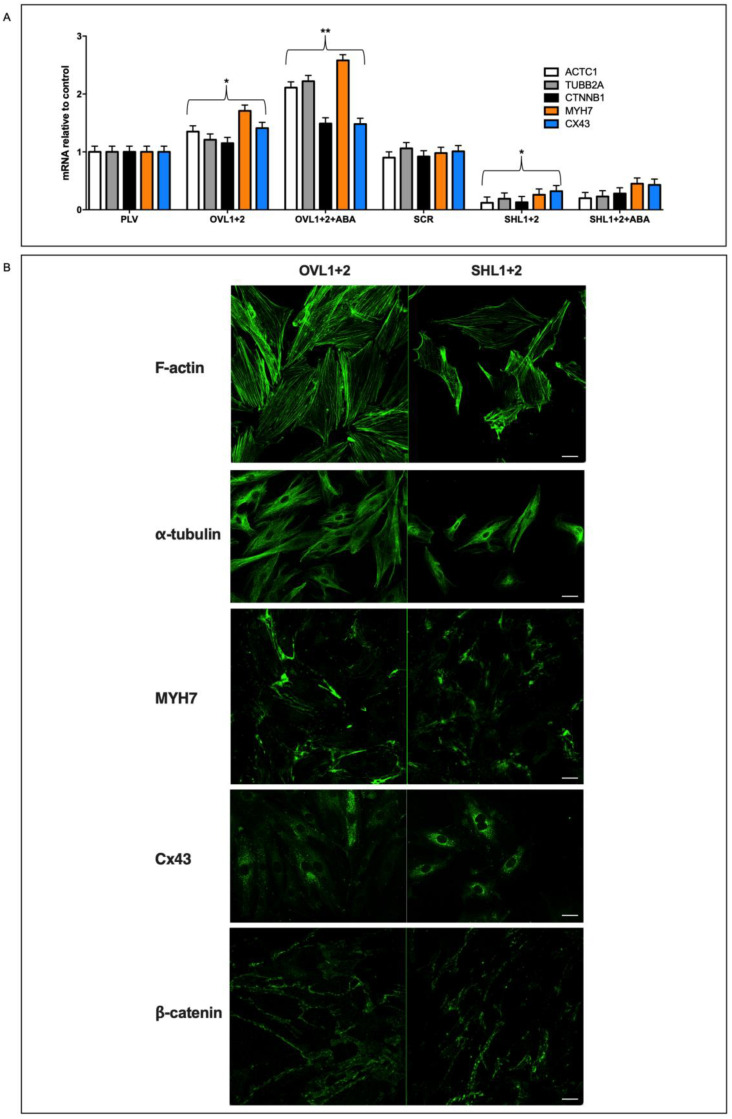
Effect of LANCL1/2 overexpression or silencing on cardiomyocyte structural proteins. (**A**) qPCR analysis of the transcription of cytoskeletal and contractile proteins (ACTC1, TUBB2A, CTNNB1, MYH7 and CX43) on OVL1+2 or SHL1+2 H9c2 cardiomyocytes and incubated in the absence or in the presence of 100 nM ABA for 4 h. Data are expressed relative to control cells (PLV or SCR). * *p* < 0.05 relative to the respective untreated control cells (PLV or SCR) and ** *p* < 0.01 relative to OVL1+2 cells by unpaired *t*-test. Results shown are the mean ± SD of 3 experiments per group. (**B**) Representative confocal microscopy images taken in at least 3 microscopic fields (scale bar: 20 µm) of the same proteins analyzed in (**A**) in OVL1+2 and SHL1+2 H9c2 cardiomyocytes.

**Figure 5 antioxidants-12-01692-f005:**
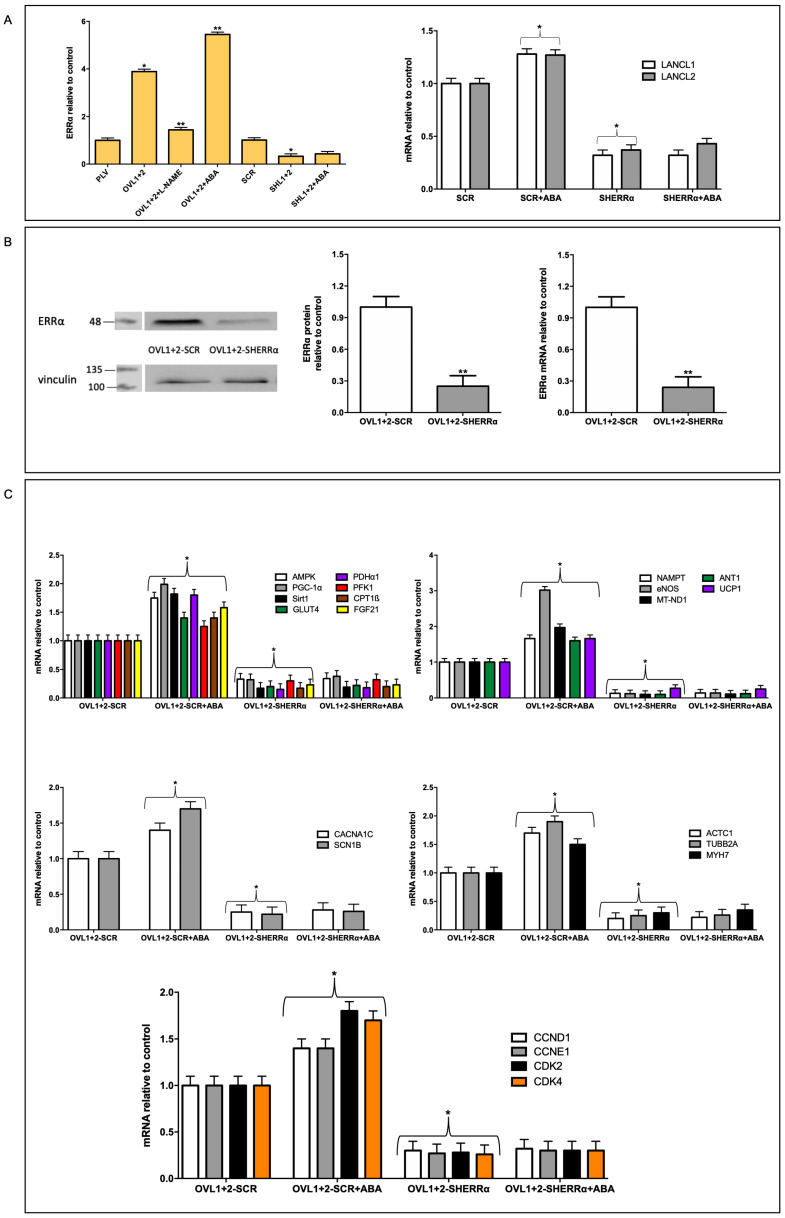
ERRα-dependent transcriptional effects on LANCL1/2-overexpressing H9c2. (**A**) Left panel, transcription of ERRα in cells overexpressing LANCL1 and LANCL2 proteins and incubated in the absence or in the presence of 100 nM ABA or 100 μM L-NAME for 4 h and in cells silenced for the expression of both LANCL1/2 proteins treated or not with 100 nM ABA for 4 h. Data are expressed relative to control cells (PLV or SCR). * *p* < 0.05 relative to the respective untreated control cells (PLV or SCR) and ** *p* < 0.01 relative to OVL1+2 cells by unpaired *t*-test. Data shown are the mean ± SD of 3 experiments per group. Right panel, mRNA levels of LANCL1 and LANCL2 in untransformed H9c2 cardiomyocytes silenced for the expression of ERRα (SHERRα, not overexpressing LANCL1/2) incubated in the absence or in the presence of 100 nM ABA for 4 h. * *p* < 0.05 relative to SCR untreated cells by unpaired *t*-test. Results shown are the mean ± SD of 3 experiments per group, with each value calculated in triplicate. (**B**) Left panel, representative Western blots of ERRα protein in ERRα-silenced, LANCL1/2-overexpressing H9c2 (OVL1+2-SHERRα) compared with control cells (OVL1+2-SCR); central panel, densitometric quantitation of the ERRα protein relative to OVL1+2-SCR; right panel, ERRα mRNA levels in OVL1+2-SHERRα cells relative to OVL1+2-SCR. Values are normalized on vinculin. ** *p* < 0.01 relative to OVL1+2-SCR control cells by unpaired *t*-test. Data shown are the mean ± SD of 3 experiments per group, with each value calculated in triplicate. (**C**) qPCR analysis of the transcription of specific genes on OVL1+2-SHERRα H9c2 and incubated in the absence or in the presence of 100 nM ABA for 4 h. Upper left panel, AMPK, PGC-1α, Sirt1, GLUT4, PDHα1, PFK1, CPT1ß AND FGF21 mRNA levels; upper right panel, NAMPT, eNOS, MT-ND1, ANT1 and UCP1 mRNA levels; central left panel, CACNA1C and SCN1B mRNA levels; central right panel, ACTC1, TUBB2A and MYH7 mRNA levels; lower panel, CCND1, CCNE1, CDK2 and CDK4 mRNA levels. Results are expressed relative to OVL1+2-SCR control cells. * *p* < 0.05 relative to OVL1+2-SCR cells by unpaired *t*-test. Results shown are the mean ± SD of 3 experiments per group, with each value calculated in triplicate.

**Figure 6 antioxidants-12-01692-f006:**
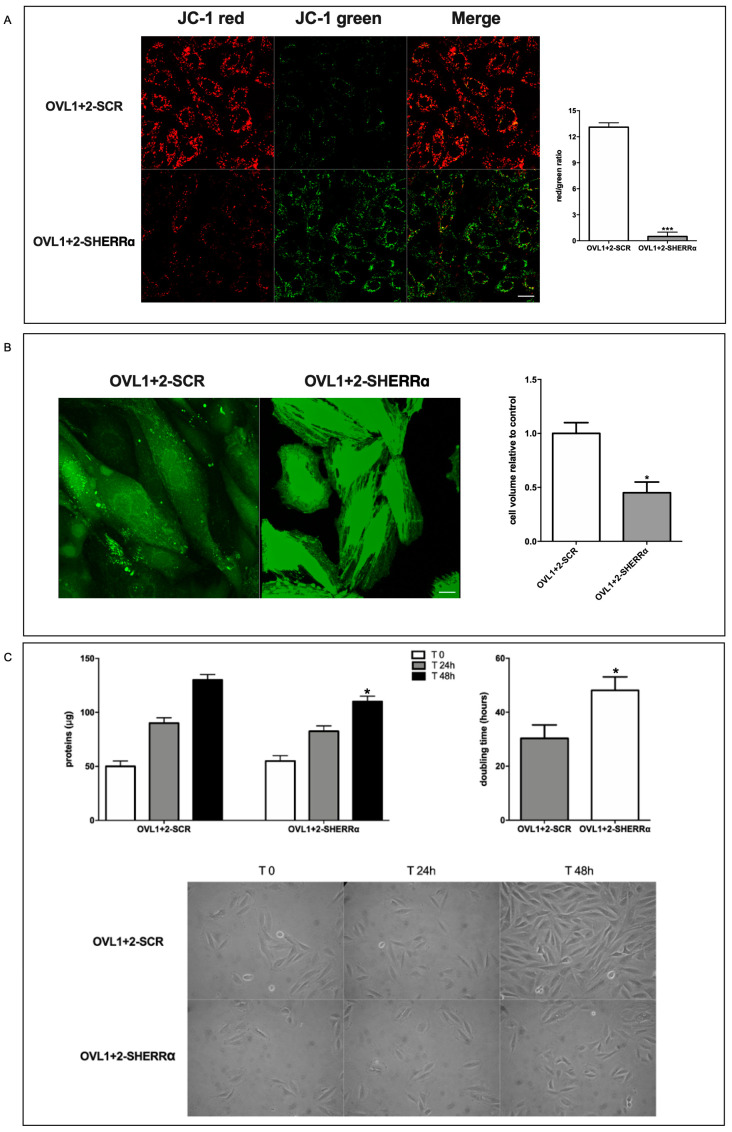
ERRα-dependent functional effects in LANCL1/2-overexpressing H9c2. (**A**) ERRα-silenced, LANCL1/2-overexpressing H9c2 were loaded with the ∆Ψ-sensitive dye JC-1. Left panel, representative confocal microscopy images; right panel, red/green fluorescence ratio calculated in at least 4 microscopic fields (scale bar: 20 µm) for each experiment. *** *p* < 0.005 relative to OVL1+2-SCR cells by unpaired *t*-test. (**B**) Cell volume was measured by confocal microscopy on calcein-loaded cells. Left panel, representative confocal microscopy images; right panel, densitometric quantitation relative to OVL1+2-SCR calculated in at least 3 microscopic fields (scale bar: 20 µm) for each experiment. * *p* < 0.05 relative to OVL1+2-SCR cells by unpaired *t*-test. (**C**) Cell protein content at time = zero (T0) and after 24 and 48 h (upper left panel) and the calculated doubling time (upper right panel). Representative photomicrographs were taken during the cell culture (lower panel, 20× magnification). * *p* < 0.05 relative to OVL1+2-SCR cells by unpaired *t*-test. Data shown are the mean ± SD of 3 experiments per group, with each value calculated in triplicate.

**Figure 7 antioxidants-12-01692-f007:**
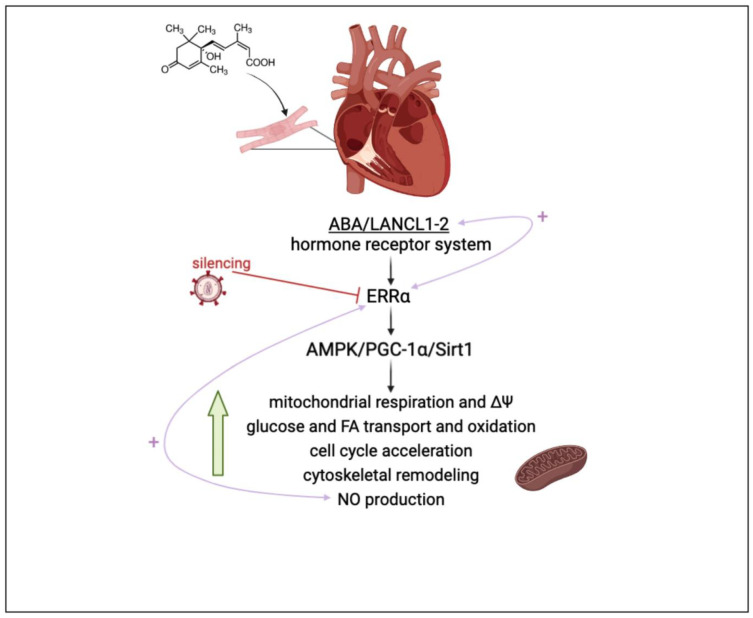
The ABA/LANCL1-2 hormone-receptor system controls cardiomyocyte fitness via NO and ERRα. LANCL1/2 overexpression activates the AMPK/PCG-1α/Sirt1 axis in H9c2 rat cardiomyocyte cells, Spinelli 2022 [5], via the orphan-receptor/transcription factor ERRα. A reciprocal transcriptional stimulation links LANCL1/2 and ERRα (right pink arrow). Downstream of this signaling axis several key functional features of H9c2 are stimulated, resulting in a higher energy availability and increased NO production, which in turn activates ERRα transcription (left pink arrow) providing a positive feedback which tends to maintain cells in this energy-proficient state.

## Data Availability

Not applicable.

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
