# Peer review of "Abscisic Acid and Its Receptors LANCL1 and LANCL2 Control Cardiomyocyte Mitochondrial Function, Expression of Contractile, Cytoskeletal and Ion Channel Proteins and Cell Proliferation via ERRα"

_antioxidants, 2023, doi:10.3390/antiox12091692_

Round 1
Reviewer 1 Report
The manuscript entitled “Abscisic acid and its receptors LANCL1 and LANCL2 control cardiomyocyte mitochondrial function, expression of contractile, cytoskeletal and ion channel proteins and cell proliferation via ERRα” by Sonia Spinelli and co-authors addresses an interesting topic related to changes in the number of mitochondria, glucose- and palmitate-dependent mitochondrial respiration, mitochondrial membrane potential, transcription of uncoupling proteins, expression of proteins involved in cytoskeletal, contractile and electrical functions, cell morphology and doubling time in rat H9c2 cardiomyocytes overexpressing, or silenced for LANCL1 and LANCL2, which are cultured in the presence or in the absence of nanomolar ABA. The authors also studied whether ERRα is involved in the signaling pathway activated by the ABA/LANCL system in the cardiomyocytes. The authors found that, as compared with double-silenced cells, LANCL1/2-overexpressing H9c2 showed: i) increased mitochondrial respiration, with higher basal and maximal respiration rates, a doubling of the spare respiratory capacity and a steeper proton gradient (ΔΨ); ii) increased fatty acid-fueled respiration rate; iii) increased NO generation; iv) reduced mitochondrial ROS content, higher expression levels of ROS-scavenging and lower levels of ROS-producing enzymes; v) increased transcription and expression of contractile and ion channel proteins; vi) improved resistance to hypoxia/reoxygenation; vii) increased proliferation rate. Based on the data obtained, the authors concluded that overexpression of LANCL1/2 significantly improved, while their combined silencing dramatically reduced key functional features of rat H9c2 cardiomyocytes, suggesting that the ABA-LANCL1/2 hormone-receptors system could control fundamental aspects of cardiomyocyte physiology via an ERRα/AMPK/PGC-1α signaling axis and ABA-mediated targeting of this axis could improve cardiac function and resilience to hypoxia.
The manuscript is interesting to read, well-organized and detailed. The Introduction section is relevant and sufficient information about the previous study findings and it is presented for readers to follow the present study rationale. The results are clear, explained with appropriate statistics. The manuscript as a whole and all its sections are well structured.
Comments
Lines 750-752: The authors mentioned here and throughout the text that “all respiratory complexes (RC) can produce ROS when electron flow is excessive”. The authors need to provide relevant literature references, or specify which specific respiratory chain complexes contribute to the formation of ROS.
Some limitations of the study may be included in the Discussion section.
Author Response
Reviewer #1
The manuscript entitled “Abscisic acid and its receptors LANCL1 and LANCL2 control cardiomyocyte mitochondrial function, expression of contractile, cytoskeletal and ion channel proteins and cell proliferation via ERRα” by Sonia Spinelli and co-authors addresses an interesting topic related to changes in the number of mitochondria, glucose- and palmitate-dependent mitochondrial respiration, mitochondrial membrane potential, transcription of uncoupling proteins, expression of proteins involved in cytoskeletal, contractile and electrical functions, cell morphology and doubling time in rat H9c2 cardiomyocytes overexpressing, or silenced for LANCL1 and LANCL2, which are cultured in the presence or in the absence of nanomolar ABA. The authors also studied whether ERRα is involved in the signaling pathway activated by the ABA/LANCL system in the cardiomyocytes. The authors found that, as compared with double-silenced cells, LANCL1/2-overexpressing H9c2 showed: i) increased mitochondrial respiration, with higher basal and maximal respiration rates, a doubling of the spare respiratory capacity and a steeper proton gradient (ΔΨ); ii) increased fatty acid-fueled respiration rate; iii) increased NO generation; iv) reduced mitochondrial ROS content, higher expression levels of ROS-scavenging and lower levels of ROS-producing enzymes; v) increased transcription and expression of contractile and ion channel proteins; vi) improved resistance to hypoxia/reoxygenation; vii) increased proliferation rate. Based on the data obtained, the authors concluded that overexpression of LANCL1/2 significantly improved, while their combined silencing dramatically reduced key functional features of rat H9c2 cardiomyocytes, suggesting that the ABA-LANCL1/2 hormone-receptors system could control fundamental aspects of cardiomyocyte physiology via an ERRα/AMPK/PGC-1α signaling axis and ABA-mediated targeting of this axis could improve cardiac function and resilience to hypoxia.
The manuscript is interesting to read, well-organized and detailed. The Introduction section is relevant and sufficient information about the previous study findings and it is presented for readers to follow the present study rationale. The results are clear, explained with appropriate statistics. The manuscript as a whole and all its sections are well structured.
Comments
Lines 750-752: The authors mentioned here and throughout the text that “all respiratory complexes (RC) can produce ROS when electron flow is excessive”. The authors need to provide relevant literature references, or specify which specific respiratory chain complexes contribute to the formation of ROS.
In addition to the three references already cited in the manuscript (45-47), several other studies have tackled the issue of ROS production at several respiratory complexes, both experimentally and via biophysical modelling.
Complexes I and III have long been known as sites of ROS production (Dröse and Brandt, 2012). At the time, complex II was not identified as a source of ROS; however, subsequent studies added complex II to the list of respiratory complexes sites of ROS production (Grivennikova 2017; Larosa and Remacle 2018; Mazat 2020).
These references (new refs. 48-51) have been added in a brief paragraph to the discussion (page 25, line 793).
Some limitations of the study may be included in the Discussion section.
A sentence regarding the need for further studies on primary culture rodent cardiomyocytes and/or on human iPSC-derived cardiomyocytes has been added in the new paragraph “Conclusions” (page 26, line 868).
I wish to thank the reviewer for the time and effort dedicated to this task!
Reviewer 2 Report
Dr. Spinelli et al. studied the effect of ABA/LANCL system on mitochondrial oxidative metabolism and structural proteins in cultured H9c2 cells. They found that overexpression of LANCL1/2 significantly increased mitochondrial number, oxidative phosphorylation, proton gradient, glucose and palmitate-dependent respiration. Overexpression of LANCL1/2 also increased transcription of uncoupling proteins, expression of proteins involved in cytoskeletal, contractile and electrical functions. In contrast, silencing of LANCL1/2 significantly decreased mitochondrial function. They also found that the LANCL1/2-dependent NO generation are mediated by the transcription factor ERRα. They concluded that the ABA-mediated LANCL1/2 activation will improve cardiac function and resilience to hypoxic condition.
It is a well-written manuscript. The reviewer only has some minor comments.
1. Statical analysis. Authors stated “Comparisons were drawn by an 242 unpaired, two-tailed Student’s t-test, if not otherwise indicated.” Please make clear if the t-test is still used in multiple comparison (more than two groups?) If yes, please justify why only t-test but not one way ANNOVA is used for statistical analysis?
2. Figure 1D ECAR panel, Did authors miss the statistical marks in this panel?
3. Discussion line 794-820. In this section, authors discussed the ROS production and proton leaking. In general, the reverse flow-induced ROS generation is more sensitive to the inner mitochondrial membrane potential. Could authors discuss effect of mitochondrial membrane potential on reverse and forward flow-induced ROS generation in this section?
Author Response
Reviewer #2
Dr. Spinelli et al. studied the effect of ABA/LANCL system on mitochondrial oxidative metabolism and structural proteins in cultured H9c2 cells. They found that overexpression of LANCL1/2 significantly increased mitochondrial number, oxidative phosphorylation, proton gradient, glucose and palmitate-dependent respiration. Overexpression of LANCL1/2 also increased transcription of uncoupling proteins, expression of proteins involved in cytoskeletal, contractile and electrical functions. In contrast, silencing of LANCL1/2 significantly decreased mitochondrial function. They also found that the LANCL1/2-dependent NO generation are mediated by the transcription factor ERRα. They concluded that the ABA-mediated LANCL1/2 activation will improve cardiac function and resilience to hypoxic condition.
It is a well-written manuscript. The reviewer only has some minor comments.
- Statical analysis. Authors stated “Comparisons were drawn by an 242 unpaired, two-tailed Student’s t-test, if not otherwise indicated.” Please make clear if the t-test is still used in multiple comparison (more than two groups?) If yes, please justify why only t-test but not one way ANNOVA is used for statistical analysis?
In fact, there are no instances when multiple comparisons were drawn.
- Figure 1D ECAR panel, Did authors miss the statistical marks in this panel?
Basal and maximal ECAR were not significantly different between silenced and overexpressing cells. The maximal ECAR was significantly higher than the basal, both in silenced and in overexpressing cells, but the statistics in the entire figure compares overexpressing and silenced cells, and in the case of the ECAR, they have similar values.
- Discussion line 794-820. In this section, authors discussed the ROS production and proton leaking. In general, the reverse flow-induced ROS generation is more sensitive to the inner mitochondrial membrane potential. Could authors discuss effect of mitochondrial membrane potential on reverse and forward flow-induced ROS generation in this section?
Indeed this is a very interesting and complex issue, that lies at the heart of mitochondrial functioning under changing oxygen and substrate availability.
Heinen et al found that under substrate conditions that allow reverse electron flow at complex II, K+ influx into the matrix through Ca2+-sensitive K+ channels (thus depolarizing the intermembrane space) accelerated the forward electron flow and concomitantly reduced ROS production in guinea pig heart mitochondria.
Ross et al. observed that a slight depolarization of ΔΨ reduces reverse electron flow-induced ROS generation at complex I in murine heart mitochondria, indicating that reverse electron flow at complex I is particularly sensitive to depolarization.
Miwa and Brand found that mild uncoupling significantly reduced ROS production from complex I on the matrix side of the membrane of insect mitochondria. Thus, several observations support the notion that a high delta psi is favorable to ROS generation, while mild proton gradient dissipation reduce ROS generation.
These references and a brief paragraph have been added to the discussion (page 25, line 809).
I wish to thank the reviewer for the time and effort dedicated to this task!